# Phenolic Metabolites from Barley in Contribution to Phenome in soil Moisture Deficit

**DOI:** 10.3390/ijms21176032

**Published:** 2020-08-21

**Authors:** Anna Piasecka, Aneta Sawikowska, Anetta Kuczyńska, Piotr Ogrodowicz, Krzysztof Mikołajczak, Paweł Krajewski, Piotr Kachlicki

**Affiliations:** 1Institute of Plant Genetics, Polish Academy of Sciences, 60-479 Poznan, Poland; apiasecka@ibch.poznan.pl (A.P.); akuc@igr.poznan.pl (A.K.); pogr@igr.poznan.pl (P.O.); kmik@igr.poznan.pl (K.M.); pkra@igr.poznan.pl (P.K.); 2Institute of Bioorganic Chemistry, Polish Academy of Sciences, 61-704 Poznan, Poland; 3Poznan University of Life Sciences, 60-624 Poznan, Poland; aneta.sawikowska@mail.up.poznan.pl

**Keywords:** drought, mass spectrometry phenotyping, metabolomics, barley

## Abstract

Eight barley varieties from Europe and Asia were subjected to moisture deficit at various development stages. At the seedling stage and the flag leaf stage combined stress was applied. The experiment was designed for visualization of the correlation between the dynamics of changes in phenolic compound profiles and the external phenome. The most significant increase of compound content in water deficiency was observed for chrysoeriol and apigenin glycoconjugates acylated with methoxylated hydroxycinnamic acids that enhanced the UV-protection effectiveness. Moreover, other good antioxidants such as derivatives of luteolin and hordatines were also induced by moisture deficit. The structural diversity of metabolites of the contents changed in response to water deficiency in barley indicates their multipath activities under stress. Plants exposed to moisture deficit at the seedling stage mobilized twice as many metabolites as plants exposed to this stress at the flag leaf stage. Specific metabolites such as methoxyhydroxycinnamic acids participated in the long-term acclimation. In addition, differences in phenolome mobilization in response to moisture deficit applied at the vegetative and generative phases were correlated with the phenotypical consequences. Observations of plant yield and biomass gave us the possibility to discuss the developmentally related consequences of moisture deficit for plants’ fitness.

## 1. Introduction

Plant resistance to moisture deficit is a variable feature depending on many factors such as edaphic, climatic and agronomic conditions, accompanying stresses as well as on the developmental phase of plants [1,2]. Plant organisms can adapt to living in stressful conditions. Adaptation to the moisture deficit is a result of changes introduced to genomes of individuals of a species in the course of evolution as a result of mutations or breeders’ efforts [3]. This allows to minimize the detrimental effect of this stress and increase the likelihood of obtaining some crop yield despite a permanent or temporary water deficit. Resistance to this stress in its physiological context is determined by “drought escape”, “dehydration avoidance” and/or “dehydration tolerance” [4]. 

Acclimation to water deficiency is in turn a noninheritable process of modification in the structures and functions of the plant build. Acclimation is often called “hardening”, whereby the plant becomes resistant to stress [5] and its possibilities depend on the genetic potential of the individual and include different mechanisms leading to changes in plant growth and physic-chemical processes. In general, acclimation is determined by the ability of the organism to accumulate protective compounds and the ability to regenerate and repair the damage caused by the stress. 

Plants utilize various mechanisms to cope with drought conditions [1]. In plants subjected to drought one of the first lines of defence is decreasing activity of photosynthetic enzymes, which helps reduce imbalance of ROS overproduction. Next, enhanced ROS can cause a cellular damage (oxidative stress). ROS can also trigger the expression of genes for ROS-dependent enzymes, like superoxide dismutase, as well as low molecular ROS scavenger-specialized secondary metabolites [6].

A significant group of nonenzymatic antioxidants involved in plant response and tolerance to moisture deficit are phenolic compounds [7]. Their high antioxidant activity depends on structural features such as the position and number of hydroxyl groups in cyclic rings. On the other hand, their methylation or glycosylation exerts the opposite effect. The spatial position of the B ring, its angle of inclination relative to the remainder of the flavonoid molecule may also affect its antioxidant properties [8]. Antioxidative phenolic compounds can protect the enzymatic complexes of the cytoplasm and chloroplasts in mesophyll and epidermal cells [9,10]. It is also suggested that these compounds form complexes with DNA and RNA in the nucleus to protect them from the damaging effects of free radicals [11]. Phenolic compounds found in the vacuole are substrates for the peroxidase in the reaction of hydrogen peroxide inactivation [12].

In moisture deficit, phenolic compounds act as energy dissipaters by absorbing the high UV radiation, therefore they prevent photo-inhibitory injuries to the photosynthetic apparatus [13]. In case of energetic imbalance under stress conditions phenolic compounds can transmit or absorb energy from other molecules ensuring continuity of photo-assimilate outlets [14]. Excess production of secondary metabolites in stress condition is costly for plants, therefore its consequences are related to the biomass and yield production [15].

Reduction of photosynthetic activity in barley plants and subsequent reactive oxygen species (ROS) overproduction are one of the long-distance consequences of water deficiency [16,17]. Disorders relating to germination, root and shoot growth and a significant reduction in the number of leaves and their transpiration surface were also noticed in barley plants subjected to water deficiency [2,18,19]. Prolonged and escalating water deficit may also result in accelerated ageing and leaf drop, and the final result is generally a significant reduction in yield [20]. At the same time, a decrease in the activity of enzymes that synthesize sugars and starch occurs, causing a decrease in grain weight and a lower content of essential components of the crop. The main goal in our metabolomic investigation was the identification of metabolites that significantly change their level under different moisture deficit with a strong focus on their structural properties. It was also of interest if such changes measured at different plant growth stages are correlated, and if they have any impact on the final plant external phenotype, as measured at their maturity. Thus, advanced bioinformatic analysis for correlation of phenolome with external phenotype is necessary to decipher their network in moisture deficit.

## 2. Results 

### 2.1. Peak Annotation

Ultra-Performance Liquid Chromatography (UPLC) profiling was executed to compare the effects of control conditions and different water regimes on phenolic compound composition. These semiquantitative analyses allowed us to observe the numbers and dynamics of changes in the content of secondary metabolites in stress. Extrapolation of the significant effects to the HPLC-MS^n^ and high resolution UPLC-MS systems were used to identify structural features of the responsive metabolites. 

The peak annotation step showed that the co-elution of different compounds occurred in some cases during the chromatographic process, despite the great care taken in optimizing the separation conditions and the use of high-resolution instruments. More than one metabolite name was assigned to a chromatographic peak in such cases. Pairs of peaks from two studied wavelengths, corresponding to the metabolites with two absorption maxima (e.g., flavonoids), were assigned the same metabolite name; in such a situation, the sum of the two peaks was treated as the observed value. The detected 104 known and still-unknown metabolites, constituting variables in the statistical analysis, are listed in Appendix A; mean metabolite concentrations are given in Appendix A and 48 additionally identified metabolites which occurred only in drought conditions are listed in Appendix A. 

### 2.2. Changes in Phenolome in Moisture Deficit

#### 2.2.1. Comparison of Effects in Three Treatments

Significant metabolites were classified by ANOVA into the following groups of response to moisture deficit: Observed effect of moisture deficit dependent on treatment (group D),Observed effect of moisture deficit dependent on the variety and treatment (VD)Observed effect of moisture deficit dependent on treatment and time of observation (treatment duration, TD)

Observed effect of moisture deficit dependent on the interaction between variety, treatment and time point of treatment duration (VTD). Table 1 shows significant differences in the dynamics of secondary metabolite profiles in barley under different treatments. The proportion of responding signals indicates a greater mobilization of phenolic metabolites in moisture deficit at the early stage of development than in the flag leaf stage and combined stress (χ^2^ test for independence using permutations, *p <* 0.001). Response to drought at metabolomic level in both treatments I and II were more dependent on the length of stress duration (effects TD) than in the treatment I+II. Those observations indicated time-specific accumulation of the metabolites. Comparison of the proportions of main effects (D) showed substantial differences in the barley response to moisture deficit at different developmental stages. Metabolomic changes in treatment II (Table 1) suggest that they might be correlated with triggering of resistance mechanisms specific for each variety at the start of the generative phase. In treatment I+II a reverse phenomenon was observed, as most of the metabolomic reactions were similar for all varieties. Below, we describe metabolites according to the type of their reaction to moisture deficit, except for not-yet-identified metabolites with a VD effect. Results and discussion will focus on the changes in the contents of the identified metabolites, however, in some cases changes of contents of unidentified compounds 54–104 (Appendix A) were also significant, e.g., No. 78 in treatment I (Figure 1). Unfortunately, lack of structural details of the compounds did not allow the drawing of conclusions about their characteristics and their impact on plants under moisture deficit. 

#### 2.2.2. Metabolites in Group D (Treatment Effect Similar for All Variety × Time Variants) 

Metabolites belonging to group D (with the exception of unidentified signals no. 61 and 65 in treatment II) were upregulated under treatment I, II and I+II, (Figure 2, Appendix A). The highest changes as a result of water deficiency were recorded for the structures of 6-*C*-[2”-*O*-glycosyl]-glucosides of flavones (signals no. 2, 6, 15–37, 45–48), which may be associated with transport of these metabolites to vacuoles and accumulation therein. In treatment I, the most significant increases of metabolites 19, 46 and 53 (identified as isoscoparin 2”-*O*-arabinoside, sinapoyl-isovitexin 2”-*O*-arabinoside, and co-eluting tricin 7-*O*-glucoside and tricin 7-*O*-[sinapoyl]-glucoside, respectively) were noticed. The contents of the two, first-mentioned metabolites also significantly increased in treatment II. Signals **3** and 35, corresponding to di- and triglycosides of apigenin (7-*O*-rhamnosylglucoside and 7-*O*-arabinosylglucoside of apigenin and 7-*O*-rhamnosylglucoside-6-*C*-glucoside of apigenin, respectively), as well as derivatives of *p*-coumaric acid 38 and 39, had the most significant increase in treatment I+II, in which acclimation processes were postulated. An increase in contents of signals 13, 38, 39, 45, 47 and 50, corresponding to *p*-coumaric, ferulic and sinapic acid derivatives, was also observed in treatment I+II. Signals 3, 6, 8, 19, 23, 30, 33, 35, were assigned as glycoconjugates of the flavones apigenin and chrysoeriol. Metabolites 13 and 19 (glucoside of hordatine B and isoscoparin 2”-*O*-arabinoside, respectively) had effects in each treatment in a variety-independent manner. 

#### 2.2.3. Metabolites in Group TD (Treatment Effect Modified by Time)

Metabolites for which effects of water deficiency were significantly different at consecutive time points were clustered into four classes A–D according to the time profile of effects (Figure 2, Appendix A). The obtained clusters were most separated for treatment I, and least separated for treatment I+II. Under treatment I, two most distant classes of compounds included metabolites with early response. Contents of compounds from the class B (red lines in Figure 2) increased in comparison to the control at the beginning of stress until the 6th day and then decreased. Glucoside of hordatine B (14), *p*-coumaroyl- and sinapoylquinic acids (39 and 50, respectively) belonged to this group of metabolites. The class A (black lines in Figure 2) of early responding metabolites showed a reverse pattern of effects (metabolites 4, 12, 40; caffeic acid derivative, hordatine B and sinapic acid derivative, respectively). The class D of metabolites (blue lines in Figure 2) included those with the observed increasing effect starting from the midpoint of the treatment duration. Among these were compounds 38 and 41 (derivatives of *p*-coumaric and sinapic acid, respectively). The dynamics of the contents of several metabolites (the class C, green lines in Figure 2, metabolites 17, 18, 29, 51) slightly differed from the control throughout the entire duration of treatment I. Compounds 17, 18, 29 belonged to the family of flavone glycoconjugates and 51 was one of isomeric sinapoylquinic acids. The profiles of the class C were similar to the profiles observed for most TD metabolites in treatment II. 

In treatment II, some of the compounds with effects changing over time (21 22, 30, 48) were flavones acylated with methoxyhydroxycinnamic acids and *p*-coumaric acid derivatives. The differences between the clusters of metabolites were not as large as those observed under treatment I. Treatment I+II was characterized by smaller variation of the effects of particular metabolites over time than in treatments I and II. Metabolites 17, 26, 48 were all flavone derivatives, moreover, compounds 26 and 48 were acylated with phenolic acids. Compounds 38, 39 and 51 reacted in a time-specific way under treatments I and II, and compound 48—under treatments II and I+II. Compound 17, isoorientin 7-*O*-glucoside, was the only one that changed significantly under all treatments, although its reaction profile was different.

#### 2.2.4. Metabolites in Group VTD (Treatment Effect Modified by Interaction Between Variety and Time)

The treatment effects for 13 metabolites were modified by variety and simultaneously by time in at least one treatment (Appendix A). Effects observed at the last time point for the identified compounds are shown in Figure 3. The most significant increase of contents as a result of treatment I was observed for metabolites eluting within the peak 8 (chrysoeriol 7-*O*-rhamnosylglucoside and luteolin 7-*O*-arabinoside). Treatment II caused lower changes for metabolites of the peak 8, in most cases not significant. In the case of metabolite 18 (isoorientin 7-*O*-rhamnosylglucoside), the pattern of dynamics was different than that for 8 and a decrease for all treatments was noticed, except for treatment II in two varieties (Maresi, Cam/B1/CI). More divergence in terms of response was noted for sinapic acid derivatives 41, 44 and 48.

### 2.3. Identification of Metabolites Induced in Moisture Deficit 

Phenolic analysis was performed using a highly sensitive and high-throughput UPLC system provided the UV profiles of phenolic compounds for all tested samples. These analyses provided a large set of data on metabolomic changes that occurred in a large number of samples. The HPLC-MS^n^ approach was used both to identify metabolites characteristic for certain groups of samples (varieties, and growth phases in which moisture deficit was applied), and especially to recognize metabolites induced under treatments in individual barley varieties.

Maresi, Cam/B1/CI and Stratus varieties were chosen for structural analysis, as they were the three most differing in their response to water deficiency and 49 metabolites not present in control plants were identified by HPLC-DAD-MS^n^ in plants subjected to treatments. For details of the identified compounds, see Appendix A (the numbering of metabolites in this Table is a continuation of the numbering from Appendix A.). 

Leaf samples from the variety Cam/B1/CI collected on the 6th day of moisture deficit applied at the seedling stage (treatment I) contained compounds 117 and 119 with similar fragmentation patterns. Fragmentation of the deprotonated ions [M-H]^-^ at *m/z* = 741 and *m/z* = 725 for 117 and 119, respectively, primarily yielded a loss of 146 amu which could correspond to rhamnose as well as to *p*-coumaroyl moieties (Figure 4a). The order of elution of these compounds from the chromatographic column indicated their highly hydrophilic properties, characteristic for glycoconjugates. Therefore, fragment 146 amu was assigned to the rhamnose moiety rather than to the phenolic acidic residue. Both 117 and 119 yielded the neutral loss of 294 amu in the MS3 corresponding to an arabinosylglucoside moiety (132 amu + 162 amu). The remaining ions at *m/z* = 301 and 285 for 117 and 119, respectively, had fragmentation patterns consistent with those observed for standards of quercetin and luteolin, respectively. Therefore, 117 and 119 were identified as *O*-rhamnoside-*O*-arabinosylglucosides of quercetin and luteolin, respectively. The presence of flavonoids substituted with three different glycosidic residues in barley was noted for the first time in barley and those compounds were the first time indicated as drought-related. Induction of flavonoids containing a combination of arabinosyl, rhamnosyl and glucosyl residues was characteristic at early stages of all the studied treatments. Similar compounds 107, 111, 120, 121, 124 were observed on the 6th day of treatment II and compounds 111 and 118 on the 6th day of treatment I+II.

Interestingly, compounds acylated with hydroxycinnamic acids containing methoxyl group(s) appeared in plants harvested at the end of treatment I. The tendency for acylated flavone accumulation was also clear in other water shortage regimes. Metabolites 133 and 134 were representatives of this group. The MS2 of these compounds in negative ionization mode revealed a loss of 192 amu and 176 amu, respectively, that corresponded to hydroxyferulic (Figure 4b) and ferulic acids. The linkage of ferulic acid with glycosides of flavones has been widely described in barley and other Poaceae plants [21,22,23], whereas glycoconjugates of hydroxyferulic acid have been identified so far in [22]. The presence of the deprotonated ion [Agly+(42–18)-H]- at m/z = 293 and the loss of the neutral fragment of 180 amu have previously been described for 2”-*O*-glucoside of isovitexin [22]. The additional loss of 162 amu in the MS2 indicated the presence of the third glycoside attached to the aglycone moiety of compound 134. Therefore, 133 and 134 were tentatively assigned as hydroxyferuloyl-isovitexin 2’’-*O*-glucoside and feruloyl-isovitexin 2’’-*O*-glucoside O-glucoside, respectively.

The fragmentation pattern of the deprotonated ion of compound 136 present in the European variety Maresi and Syrian Cam/B1/CI was characterized by [M-H-368]^-^ similar to that observed for the 7-*O*-[6”-sinapoyl]-glycoside of flavone [22,23]. The loss of 150 amu in the MS3 and the detection of an ion at *m/z* = 293 [Agly+(42–18)-H]^-^ after their fragmentation indicated the structure of 2”-*O*-arabinoside of isovitexin, similar to compounds 133 and 134. Thus, the compound was ascribed as 7-*O*-[6”-sinapoyl]-glycoside 2”-*O*-arabinoside of isovitexin, in which the acyl moiety bound to the glycosidic residue in a different manner than in 133 and 134.

Compounds 152 and 153 were observed in plants of the Syrian variety Cam/B1/CI at the end of moisture deficit at the seedling stage (Figure 4c). The loss of 44 amu in both compounds was typical for fragmentation of the [M-H]^-^ ions of malonylated glycosides of flavonoids in negative ionization and it corresponded to the rupture of the carboxylic moiety [22,24]. The rest of the substituent remained on the glucose moiety as a ketene and its release gave a neutral loss of 204 amu (42 amu + 162 amu) in compound 153. The loss of 162 amu observed in the case of 152 corresponded to a typical fragment of *C*-glycosides (120 amu) bearing the ketene moiety. The fragmentation pattern of the ion at *m/z* = 271 observed for 153 was similar to that of the standard of naringenin. Hence, it could be concluded that 152 and 153 were malonylated 6-*C*-glucoside of apigenin (isovitexin malonylated) and malonylated 7-*O*-glucoside of naringenin, respectively. Compound 153 has been described for the first time in barley.

Flavonoid glycoconjugates acylated with two hydroxycinnamic acid moieties were induced by treatment I+II in leaves of Maresi variety plants. The MS^n^ mass spectrum of compound 147 in the negative ionization mode indicated the structure 2”-*O*-glucoside of isovitexin, similar to compounds 133 and 134. Location of the acyl groups in compound 147 explained the double track of the fragmentation of the ion [M-H]^-^ (Figure 5). The loss of two fragments with a mass of 206 amu in the MS2 and 224 amu in the MS3 indicated the presence of two sinapoyl groups bound by ester linkages with the hydroxyl groups of the glycosidic substituents. The predominant ions in the MS2 [M-H-206]^-^ and [M-H-206–180]^-^ corresponded to the sequential losses of sinapic acid and 2”-*O*-glucose. One of these acyls was linked to the 2”-*O*-glucose, presumably at the C6” hydroxyl. The secondary loss of a fragment of 326 amu corresponded to the second sinapic acid moiety bound to a *C*-glucoside (206 amu + 120 amu). The MS3 fragmentation of 147 in the negative ionization indicated the secondary loss of the whole acid molecule as the fragment of 224 amu, followed by fragmentation of 6-*C*-glucoside-2”-*O*-glucoside. Therefore, the 147 was tentatively described as disinapoyl-isovitexin 2”-*O*-glucoside. To our knowledge, doubly acylated flavonoids, such as compounds 146 and 147, have not been reported in barley so far.

### 2.4. Moisture Deficit and Phenotypic Assessment at Maturity

All the observed phenotypic traits, with the exception of 1000-grain weight (F5), were affected by at least one of the treatments (Table 2 and Appendix A). The largest number of traits with variety-specific effects were observed under treatment I (5 traits), and the smallest (3 traits) under treatment I+II, and referred mainly to tillering traits F1 and F2 and lateral spikes traits F13–F15; less frequent inter-varietal differences existed for other groups of traits. 

Significant, but similar to each other water deficit effects (D) were recorded for all varieties for six traits: grain weight per plant (F4), length of main (F6) as well as lateral stems (F11), number of spikelets per main spike (F8), number of grains on main spike (F9) and length of lateral spikes (F12). The mean effects of water deficiency were negative in all cases (Figure 6). A similar decrease in phenotype values was observed for treatments I and I+II, and this was bigger than for treatment II. 

Differences in treatment effects among varieties (the VD effect) were significant in at least one treatment for eight traits: F1, F2, F3, F7, F10, F13, F14 and F15 (Table 2, Figure 7). The number of tillers per plant (F1) increased in water shortage in some varieties, especially under treatment II in Lubuski, Maresi, Sebastian and Georgie (European varieties). The number of productive tillers per plant (F2) decreased in all varieties, with the lowest effects in Harmal and MDingo. A similar situation was observed for straw weight per plant (F3). The reduction of the main spike length (F7) was large for Stratus and Harmal and much smaller for the rest of the varieties, especially under treatment II. Inter-varietal differences in grain weight on main spike reduction (F10) were significant under treatment I+II, which was caused mainly by a lack of effect in Lubuski and Cam/B1/CI, and a large reduction in Stratus and Harmal. For three traits characterizing lateral spikes, i.e., number of spikelets (F13), number of grains (F14) and grain weight (F15), the treatment effects were variety-specific under early and late treatment, which can be explained mostly by large reductions for Harmal and MDingo. Linking the above-mentioned observed effects of treatment on main yield-related traits (F2, F10, F13, F14 and F15), we can say that although the reduction of grain weight per plant (F4) was similar for all varieties, it had different main causes: reduction in number of productive tillers (F2) in European varieties and Cam/B1/CI, reduction in grain weight on main spike (F10) in Stratus and Harmal, and reduction in lateral spike productivity (F13, F14 and F15) in Asian varieties, Harmal and MDingo.

A synthetic picture of plant reaction to moisture deficit can be obtained from the multivariate analysis performed for traits for which variety-specific effects were observed (Figure 8). Under treatment I, varieties MDingo, Harmal and Stratus appeared to be the most distant from the rest of varieties. Separation of MDingo and Harmal was mostly caused by a large reduction of the correlated lateral spike characteristics (F13, F14 and F15) and a relatively small reduction of F2 and F3, whereas Stratus was distinct mostly due to a large reduction of main spike traits (F10 and F7). A similar relation was observed under treatment II: Harmal and MDingo were separated mostly due to the relatively small effects of treatment on the number of productive tillers (F2) and the large effect on correlated lateral spike traits, while Stratus was separated due to its experiencing the lowest treatment effect on straw weight (F3). The rest of the varieties formed a relatively homogeneous group. Under treatment I+II, the differentiation of varieties shown on the biplot was lower, with some separation of Harmal related to a low reduction of the number of productive tillers (F2). A lack of specifically correlated trait effects was also observed.

### 2.5. Correlation between Phenotypic and Metabolomic Traits in Moisture Deficit

To study the relationships between plant phenotypic response to moisture deficit and metabolomics, Pearson correlation coefficients of variety-specific treatment effects were computed (Appendix A) at two significance levels: *p* < 0.01 and *p* < 0.05. The highest correlations (*p* < 0.01) were observed under treatment I (Appendix A); due to a low number of effects being correlated (8), just a few were significant. Out of the identified metabolites, only metabolite no. 44 (sinapic acid-glucoside) had significant correlations with phenotypic characteristics F3 (0.80) and F15 (−0.94): the levels of this metabolite increased markedly in Harmal and MDingo, which was accompanied by a relatively small decrease in straw weight and a relatively large decrease in grain weight on lateral spikes in these varieties. Another sinapoyl derivative, metabolite 48 identified as sinapoyl-isovitexin 2’’-*O*-glucoside, was highly correlated with traits related to lateral stems and spikes (traits F13, F14 and F15). The correlation between metabolite 48 and other phenotypic traits related to tillers and biomass production F2 and F3 (number of productive tillers per plant and straw weight per plant, respectively) was at the same level as for F13, F14 and F15. Metabolite **8** (assigned for two co-eluted compounds chrysoeriol 7-*O*-rhamnosylglucoside and luteolin 7-*O*-arabinosylglucoside) had the highest correlation with the phenotypic trait F13 and slightly smaller with F14 and F15. Seven unidentified metabolites were also correlated with phenotypic characters under drought applied at the seedling stage. Among them, metabolites 54 and 82 had significant (at the *p* < 0.05 level) correlations with F13 (number of spikelets per lateral spike). The lowest correlation of all metabolites was observed for traits F1 and F10 (number of tillers per plant and grain weight on main spike, respectively). On the other hand, traits of lateral stems and spikes: F13, F14 and F15 were the most correlated with secondary metabolism in barley.

## 3. Discussion 

### 3.1. Design and Data Analysis

Literature describing plant features in moisture deficit usually concerns one period of stress at a subjectively selected developmental phase. Reactions to stress in plants are developmentally-related and spatio-temporal changes in metabolic processes occur [2]. Therefore, we applied three different treatments at different plant growth stages to study the metabolomic responses and the phenotypic consequences of moisture deficit at vegetative and generative phases. The three-leaf stage is critical for the beginning of tillering, and the flag leaf stage is a preparation step for inflorescence formation, when the spikes and the number of spikelets per spike are determined, as well as when the beginnings of flowers and stamens, stigma and the embryo sac are formed [25]. In our experiments the treatment starting at the flag leaf stage covered 14 days in which the plants initiated their generative phase. In this time, the flag leaf vitality and its photosynthetic activity were critically important. 

A slight decrease in water availability in the soil at the juvenile stage may accelerate the development of the root system resulting in increased resistance to moisture deficit in the later period of plant development [26]. Thus, treatment I+II, consisting of the first period of water deficiency at the three-leaf stage and the second at the flag leaf stage was applied to observe a long-term acclimation processes in barley.

The application of several treatments, observational time points, and the proper replication of biological samples considerably increased the number of the analyzed chromatographic profiles. Even after the selection of only two UV wavelengths for a detailed data analysis, the total number of compared chromatograms exceeded 500, with a number of retention time points of more than 10000. To our knowledge, the analysis of such a large set of chromatographic data from the UPLC-UV protocol has been considered so far only in [21] (for large analyses, see, e.g., [27,28]). On the other hand, LC-MS systems for metabolomics are still expensive, which limits their availability to most researchers. Operation on LC-MS requires extensive knowledge and data processing is complicated and requires extensive theoretical preparation [29] 

### 3.2. Phenolic Metabolites in Moisture Deficit

Significant differences in phenolic metabolite profiles between the control and stressed plants were observed suggesting their important role in barley response to water deficit. Similar phenomenon was recently revealed in *Arabidopsis thaliana* leading to a notion that these compounds enhance plant tolerance to water deficit [30]. The differences in phenolome response between critical phases of development (Table 1.) reflect the dynamic and plasticity of their biosynthetic pathway. It is known that glucosyltransferases responsible for diversity in flavonoids glycosides in barley are regiospecific and they show different activities in younger and older tissues, contributing to the differentiation of the glucosylation pattern at different stages of plant development [31]. Significant increases of levels of flavone glycosides as well as their acylated forms, glycosides of hordatine B and hydroxycinnamic acid, were observed as a general phenolomic trend in barley response to water deficit (Figure 1).

A high number of phenolic metabolites changes in water deficiency were shared by all the studied varieties and only a few of the changes were specific to a particular variety (Figure 3). This may suggest a similarity in the dynamics of phenolic metabolite contents in each variety and indirectly implicates similarities in their functions. The most numerous variety-dependent effects were noted in metabolites in treatment I. Interestingly, physiological and transcription factor analysis performed in growth chambers indicated large, stress-level-dependent differences between the barley varieties Georgie, Lubuski, Maresi, Sebastian, Express, Saida and Cam/B1//CI 08887/CI 05761 in this stage [16].

Time-dependent effects for most metabolites were changed in a nonlinear manner (Figure 2.). Interestingly, most of the TD metabolites were derivatives of hydroxycinnamic acids, precursors of all the phenylpropanoid family, whereas the effects of flavones in comparison to the effects of hydroxycinnamic acids were less severe (Appendix A). 

The reorganization in the phenolic metabolism in barley under treatment I at the early developmental stage depends mainly on the mobilization of compounds from downstream of the phenolic metabolite biosynthesis pathway. The early increase in time-dependent metabolites in the class B (red lines in Figure 2.) may be caused by intensified biosynthesis in the initial phase of moisture deficit and their decrease after 6 days of the stress duration may be a result of their usage in defense processes launched in the prolonged water deficiency. Metabolites from the class A (black lines in Figure 2.) (characterized with an initial decrease and further increase) may be intensively used in the first phase of treatment. The relationship between the decrease in hordatine B (from group A) effects and simultaneous increase in the effects of its glucoside (from the class B) in treatment I may be related to using the existing pool of this compound in the further synthesis of the glucoside. The subsequent increase in hordatine B in the prolonged moisture deficit could be the effect of the permanent synthesis of the compound with a simultaneous stopping of the glycosylation reaction. Hordatines were previously reported as antifungal agents [32,33], but our report as well as [21] also showed the drought-related function for these compounds for the first time.

Particular attention should be paid to a substantial number of derivatives of *C*-glycosides of flavones apigenin, chrysoeriol and luteolin among the metabolites synthesized de novo in the water deficit conditions (Appendix A.). The drought-related flavonoids substituted with three different glycosidic residues were identified for the first time in barley. Glycosylation of flavonoids is required for their transport and accumulation in the vacuoles [9]. In treatment I+II, most metabolites were mobilized earlier than in the other regimes and their contents gradually decreased throughout the stress duration (Figure 2.). The first period of moisture deficit caused an accumulation of metabolites in glycosylated form which was the basis of a rapid reaction during the second stress applied at a later stage. The rapid response may be a result of acclimation processes and sustained as long-term resistance. Similar results have been observed for acclimation to moisture deficit in *A. thaliana* [34]. An increase in the expression of genes for chlorogenic acid and flavonoid biosynthesis has been noticed in the acclimation of coffee [35] and loblolly pine [36], respectively.

Luteolin and quercetin, in which the hydroxyl groups are in the *ortho* position, can effectively scavenge or peroxidate the ROS overproduced due to photo-oxidation under water deficit. It has been also shown that the chloroplast-located flavonoids are crucial in the inactivation of H_2_O_2_ [37]. Thus, the luteolin and quercetin derivatives 111, 116–121 and 140 induced in barley by treatments and 15–18 and 37 may participate in the reduction of H_2_O_2_. 

Compounds with a relatively low antioxidant capacity, mainly mono-hydroxyl-B-ring flavones apigenin, chrysoeriol and tricin, acylated with ferulic, hydroxyferulic and sinapic acids: 129–131, 133, 136, 138, 139, 141, 142, 145–148, appeared in barley plants during all applied treatments (Appendix A). It is noteworthy that most metabolites whose contents significantly changed in all treatments, have methoxyl groups in their structures, either directly on the aglycones or in the acyl moieties (4, 5, 6, 12–14, 19, 21, 22, 28-30, 38–41, 44–48, 50, 51, 53). In addition, metabolites with double acylation (145–147) were also induced in drought. The methoxyl groups present in metabolites reduce the antioxidant properties of molecules by blocking their free hydroxyls [10]. At the same time, they modify the hydrophobicity of the molecule, which can facilitate movement to other cell compartments. However, it seems that the shift of the UV absorption maximum towards longer wavelengths is the most important effect of the methylation occurring in drought as was previously suggested [21]. In addition, acylation of flavonoids significantly increases the absorption of UV-B radiation in comparison to the respective glycosides [38]. We suggest that the expansions of the range of absorbed UV light followed by increasing the intensity of absorption may be a protection strategy against excessive UV radiation upon drought-related photosystem failure in barley. 

There were a few metabolites which accumulated in each treatment. Similar effect of increased accumulation in each variety was observed for the glucoside of hordatine B and isoscoparin 2”-*O*-arabinoside, whereas contents of 7-*O*-glucoside of isoorientin changed time-dependently and their content approached the content in the control plants at the end of all treatments. Hordatines are hydroxycinnamic acid amide (HCAAs) that are widely distributed in higher plants but their specific structures with polyamine agmatine conjugated to *p*-coumaroyl or feruloyl component are unique to barley [39]. The regulation of developmental processes and cell division by polyamines and their acylated forms, as well as their antioxidative properties, have been described in many plants (reviewed by [40]). The role of HCAA in barley under different abiotic stresses was previously postulated [21,41]. The importance of HCAA in plant nonenzymatic free radical quenching in photosynthetic centers has also been suggested [42]. Another postulated role for hordatines in barley is related to cell wall stabilization in biotic stresses [43]. On the other hand, the acylated polyamines in barley have been observed to be jasmonate- and ABA-inducible under osmotic stress [44,45] and their participation in UV-attenuation due to the presence of a ferulic acid moiety in the structure is possible [13]. 

The structural analysis of barley phenolome indicated its potential diverse activities in moisture deficit. These findings are in agreement with earlier conclusions regarding the multiple roles of phenolic compounds in abiotic stresses with highlighting the attenuation of stress-related ROS and UV imbalance [46]. Nevertheless, 48 metabolites in barley leaves were detected as synthesized de novo in drought. Among them flavonoids containing acylation with methoxyl-rich acids (ferulic, hydroxyferulic and sinapic acids) were observed. In addition, flavonoids with three different glycosidic substitutions: glucose, rhamnose and arabinose were also primarily identified. Presence of hexose simultaneously with deoxyhexose and pentose indicated on importance of glycosylation diversity during drought in barley. Malonylation of flavonoids is common in leguminous plants and is related to response to different biotic and abiotic stresses including drought [24,47]. Presence of malonylated flavonoids in barley has been previously demonstrated [22]. Our results indicate on relation of this acylation with drought. LC-MS methods enabled for detection of new, drought-related metabolites 108, 117, 119, 124, 135, 146, 144, 146, 147, 148 and 153 which have not been previously described in barley. Further NMR analysis should be done for confirmation of these structures due to impossibilities to resolve some structural elements in these compounds as for example differentiation between glucose and galactose or place of glycosylation and acylation in some cases.

### 3.3. Phenotype and Moisture Deficit

Maintaining plant reproduction ability in stress conditions is often accompanied by changes in the phenotype [2,48]. Cam/B1/CI, MDingo and Harmal are varieties of barley from Syria and are moderately tolerant to water deficit. Maresi, Lubuski and Stratus are of European origin and we observed that water deficit affected their biomass much more than it was in the case of the Syrian varieties. Our observations of post-harvest phenotypic traits differ from those presented by [16] that were based on the biomass measurements at the three-leaf stage. Differences of effects caused by the water deficiency applied at the seedling and mature plants stages support the observation that, although the reduction of grain weight per plant was not significantly different for the Asian and European varieties, the main sources of this reduction were different: the decreased lateral spike productivity in the former and the reduction in the number of productive tillers in the latter case.

The literature indicates that the consequences of moisture deficit are the most drastic for stress applied at generative phases [49] and that late stress causes more effects on phenotype than stress at the three-leaf stage [18,50]. The analyzed varieties already exhibited sensitivity to water deficiency at the early developmental stage in our study. In fact, the effects of the early treatment were usually larger than those of late treatments. This observation may be related to additional changes in environmental conditions not controlled in the greenhouse. 

We observed similar phenotypic effects of treatment I (applied at the three-leaf stage only) and of the treatment I+II (applied at both stages). This may suggest that acclimation processes after the end of the first period of moisture deficit took place and the second period of stress influenced the plants to a smaller extent. Therefore, mainly the treatment I significantly modified the yield-related traits.

As regards the particular phenotypic effects, reduction in features related to spikes, and main and lateral stem productivity have been observed in barley [25]. The tendency to produce more tillers in all treatments is visible in European varieties, but the cost of this is a lower number of productive tillers (Figure 8). On the other hand, in the Syrian varieties’ plant outgrowth was not increased to diminish stress exposure, but the productivity of tillers decreased. It emerged that both strategies led to a similar yield reduction. Reduction of the main spike productivity was smaller in the European varieties 

### 3.4. Correlation between Phenotype and Metabolome in Moisture Deficit

The influence of metabolomic reprogramming caused by drought on phenotype at maturity manifests by correlation of methyl-containing metabolites like sinapic acid glucoside (metabolite 44) and sinapoylated glycoside of apigenin (metabolite 48) (Appendix A) and diversity in glycosyletd forms of flavonoids. Sinapic acid as well as ferulic acid esters are integral components of cell walls [13,51]. The observed correlation between sinapic acid derivatives contents and phenotypic features concerning lateral stem yield and straw yield may indicate the significance of the compound in cell wall strengthening in barley. Similar results in wheat have shown that cell-wall-bound phenolics in moisture deficit influence the activity of the photosynthetic apparatus, plant biomass and grain yield and may serve as mechanical cell wall stabilizers, preventing water loss from the apoplast [51].

Recent research shows the relationship between auxin and flavonoid concentrations, and morphogenic responses in abiotic stresses (for review, see [46]). Since it is known that a flavonoid-based mechanism regulates auxiliary bud outgrowth probably by triggering auxin movements from main stems to lateral stems, the changes in whole-plant phenotype in barley may be correlated with the regulatory (modulatory) function of flavonoids in moisture deficit. A significant increase in apigenin derivatives with mono-hydroxyl B-ring structures in treatments may be related to auxin transport from main to lateral stems. The relationships between the accumulation of flavonoids and maintenance of biomass growth under water deficit conditions have been observed for clover (*Trifolium repens* L.) [52] and recently for *Brachypodium* spp. [53]. However, a doubt regarding the importance of flavonoids in shoot branching by showing that the mutant of *A. thaliana* with severely reduced flavonoid content had only modest changes in auxin transport and did not exhibit a substantial branching phenotype was also expressed [54]. 

The final conclusions can be that moisture deficit at the three-leaf stage caused more variety-specific changes, both in metabolome and in phenotypic properties, than the two other treatments. The acclimation of plant to drought (treatment I+II) resulted in faster and stronger response to drought at the generative phase but the phenotypic traits measured at maturity were triggered in a similar way as after drought applied at the seedling stage. The drought applied at the flag leaf stage had the mildest effect at phenotypic and metabolomic level. This confirms the importance of water deficiency at early developmental stage for phenotype output at maturity.

The metabolomic reprograming during drought was mainly related to the increased accumulation of compounds containing methyl groups. It was manifested by an increased accumulation of ferulic, hydroxyferulic and sinapic acid derivatives. On the other hand, significant diversity of glycosidic forms of flavonoids can be also observed in drought. Compounds which were previously not related to drought such as hordatines and malonylated flavonoids also significantly changed accumulation. The increased accumulation of certain specialized metabolites has an impact on phenotypic traits, mainly related to the lateral stems and spikes whereas it had a lesser extent for the biomass and yield production.

## 4. Materials and Methods

### 4.1. Plant Material Cultivation

Eight spring barley (*Hordeum vulgare* L.) varieties were used in the experiment: European varieties Georgie (bred in England), Maresi (Germany), Sebastian (Denmark), Lubuski and Stratus (Poland) and Syrian accessions Cam/B1//CI08887/CI05761 (shortly described as Cam/B1/CI), Harmal and MaresDingo/DeirAlla 106 (described as MDingo). All varieties are stored in Institute of Plant Genetics, Polish Academy of Sciences. Details about the experiment preparation, plants cultivation, stress application and biometric measurements were reported previously [21].

Briefly, plants were grown in a greenhouse in pots filled with loam-sandy soil (7:2 *w*/*w*). Soil moisture was established at pF 2.2–3.0 and pF 3.4–3.6 for control and water shortage, respectively, according to a previously developed pF curve [19]. Soil moisture was controlled gravimetrically by daily weighing of each pot and, additionally, volumetrically (if necessary) using the FOM/mts device [55].

Four treatments were applied as follows: control—pF 2.2–3.0 of soil moisture during entire vegetation season; water deficiency I (treatment I)—soil moisture in the pots was decreased to pF 3.4–3.6 when seedlings were at the three-leaf stage (13 BBCH scale) and the moisture level was achieved within 48 h and maintained for 10 days; water deficiency II (treatment II)—the decrease in soil moisture to pF 3.4–3.6 was started when plants were at the flag leaf stage (37 BBCH scale) and maintained for 14 days covering the first steps of the generative phase; water deficiency I+II (treatment I+II): plants subjected to treatment I, then rehydrated and maintained in control conditions until they reached the flag leaf stage and then subjected to treatment II.

Pots were arranged in complete randomized blocks with 6 replications: 2 for phenotyping and 4 for metabolome profiling. Of note, an average from ten plants from a one pot was expressed as results from phenotypic traits of each treatment. The phenotypic traits that were measured at maturity are listed in Table 2. Straw weight per plant, grain weight per plant, 1000-grain weight were measured after drying the plant material. The distributions of the traits were symmetric and close to a normal distribution.

Leaf samples for metabolomic analyses were harvested from plants subjected to treatments I, II and I+II after 3, 6 and 10 days of moisture deficit, and in treatment I+II after 1 day. On the same days, leaves from control plants were collected. The harvested material was immediately frozen in liquid nitrogen and stored at –80 °C until metabolite extraction and instrumental analysis. 

### 4.2. Phenolic Compounds Extraction from Plant Material 

All reagents and solvents for extraction, as well as UPLC and HPLC-MS analyses (methanol, acetonitrile, formic acid, dimethylsulfoxide, polyamide and chloroform), were from Sigma-Aldrich (Poznań, Poland), and ultrapure water was obtained from a Millipore Direct Q3 device. Standards of flavonoids (apigenin, apigenin-7-*O*-glucoside, apigenin-6-*C*-glucoside, apigenin-8-*C*-glucoside, isorhamnetin, luteolin, luteolin 7-*O*-glucoside, luteolin 4′-*O*-glucoside, luteolin-8-*C*-glucoside, luteolin-6-*C*-glucoside, quercetin 3-*O*-galactoside, quercetin, rutin, isorhamnetin, isorhamnetin 7-*O*-glucoside) were purchased from Extrasynthese (Genay, France). 

Extracts were prepared according to [21]. Weighted samples of frozen leaves (about 100 mg) were placed in Eppendorf tubes with 1.4 mL of 80% methanol and the internal standard, homogenized using a ball mill (MM 400, Retsch, Haan, Germany) and subsequently placed in an ultrasonic bath for 30 min and centrifuged (11,000× *g*) for 10 min. Apigenin (5 µL of 1 mg/mL solution in dimethylsulfoxide) was used as the internal standard in extracts prepared for the UPLC analyses. Samples of the secondary metabolite extract were directly subjected to HPLC-MS analysis and, prior to the UPLC analysis, 200 µL were diluted with 80% methanol (800 µL).

### 4.3. Phenolic Compounds Profiling with UPLC-PDA 

The summarizing flowchart of molecular and chemometric analyses described in this paper is presented in Appendix A. Profiling of secondary metabolites was performed with UPLC (ultra - performanceliquid chromatography) using the Acquity system (Waters, Milford, Connecticut, USA) with a photodiode-array detector PDAeλ. The studies were carried out as previously described [21]. The signals on UPLC chromatograms were annotated by extrapolation to compounds identified in two complementary LC-MS (liquid chromatography—mass spectrometry) systems: HPLC-MSn and UPLC-HR/AM-MS/MS (described in details in [21,22]). The individual compounds were identified via comparison of the exact molecular masses, mass spectra and retention times to those of standard compounds, databases available online and literature data [21,22,23,56]. Data for 104 identified metabolites were submitted to the Metabolights database [57], identifier: MTBLS52 released 20.11.2014 (available at: http://www.ebi.ac.uk/metabolights/MTBLS52). The detected metabolites are listed in Appendix A.

### 4.4. Chromatographic Data Pre-Processing

The UPLC-PDA chromatographic profiles and resulting data were processed according to the flowchart presented in Figure 9 and Appendix A. Briefly, the raw data were normalized by dividing every single chromatogram value by the mass of the appropriate extracted leaf sample. As the chromatographic baselines can negatively affect the alignment and ultimately the results of the data analysis, they were removed without their explicit estimation using the numerical differentiation according to [58]. Retention time alignment was performed using correlation optimized warping (COW) [58], using the correlation coefficient as the similarity measure of two peaks. 

Computations were implemented in R scripts (available at: https://github.com/sawikowska/POLAPGEN-2011). 

Following the alignment, peaks were detected in individual chromatograms using profiles of the second derivative smoothed by a cubic smoothing spline (with 520 knots). Some detected individual peaks were considered to be irrelevant and were removed from the subsequent analysis. The irrelevant individual peaks were characterized either by a length of 1 (covering only one retention time point), or by a low intensity value within the peak after differentiation and warping (below a tolerance of 0.0005). Peaks common for all chromatograms were built as the sums of the individual ones (in the sense of interval addition). The minimum number of chromatograms required for the definition of a common peak, *c*_1_, was chosen as the number for which the desired (identified) compounds were not lost, and, on the other hand, the number of observations within it was appropriate for the statistical analysis applied to the data. The peak detection in the COW aligned chromatograms revealed 84 and 83 peaks for wavelengths 280 nm and 330 nm, respectively. 

### 4.5. Statistical Analysis

Results from the chromatographic data preprocessing (transformed by log_2_(10^9^×)) and phenotypic data were analyzed (each trait independently) by analysis of variance based on the REML (restricted maximum likelihood) algorithm. The significance of variation sources was checked by tests based on the *F* approximation of the Wald statistic. For phenotypic traits measured at maturity, the ANOVA model included (fixed) effects of variety (V), kind of treatment (D) and the VD interaction; significant effects were declared at *p* < 0.01. For metabolites, the ANOVA model included (fixed) effects of variety (V), time of observation (T), treatment (D), and effects of all possible interactions. To correct for simultaneous testing, independently in each group V, T, D, VT, VD, TD, VTD of effects, the significant effects were selected using the Bonferroni approximation to achieve the family-wise error rate (FWER) < 0.05. Only D, VD, TD and VTD effects are discussed in the interpretation of the results, as these are the ones representing effects of moisture deficit on the metabolite level and modifications of these effects by variety, time and interaction between variety and time. Treatment effects for individual experimental combinations were computed as (concentration in water deficiency—concentration in control), and mean factor effects were computed in a similar way from the appropriate average values. Time profiles of effects were clustered (in the multivariate sense, considering measurements at different time points as separate variables) using the maximum inter-group sum of squares criterion with the number of groups selected arbitrarily as four. Mahalanobis distances between groups were computed. Biplots based on treatment effects were used for the visualization of the similarities between varieties and correlations of traits. All statistical analyses were conducted in Genstat 14 (VSN Int.).

## Figures and Tables

**Figure 1 ijms-21-06032-f001:**
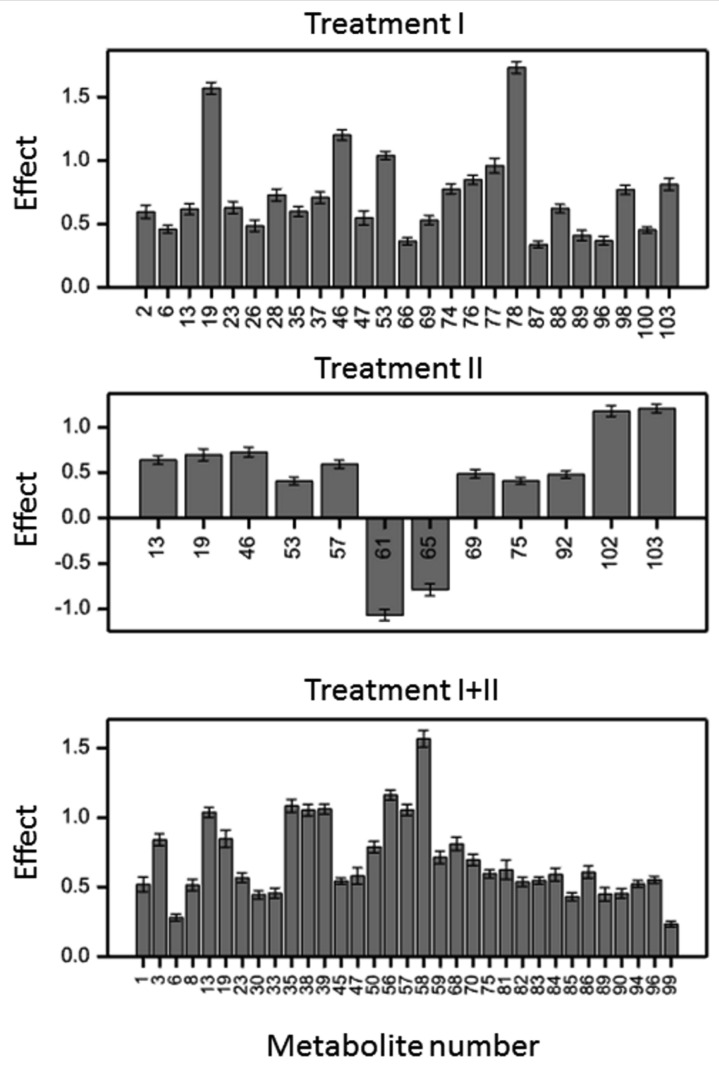
Effects of water deficiency for metabolites that reacted in a similar way in all varieties and at all time points, under treatments I, II and I+II.

**Figure 2 ijms-21-06032-f002:**
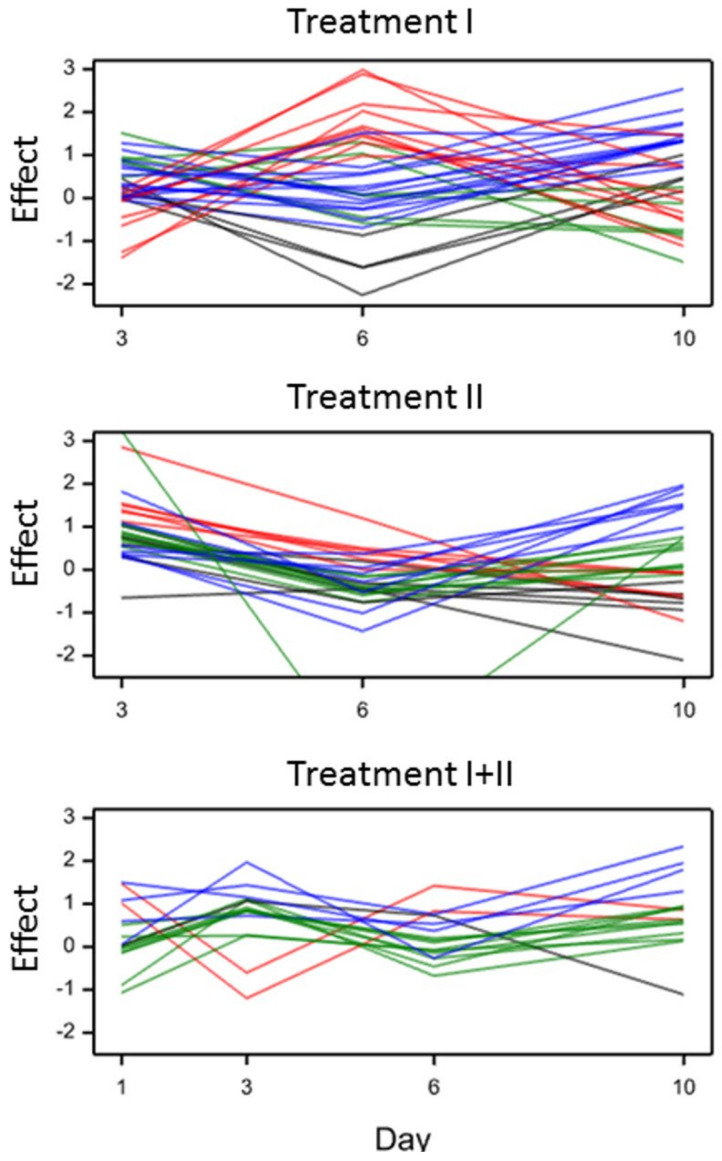
Time profiles of effects of water deficiency (averaged over varieties) for metabolites characterized by effects significantly modified by time (treatment duration (TD) effects); clusters of metabolites with different profiles: A—black, B—red, C—green, D—blue.

**Figure 3 ijms-21-06032-f003:**
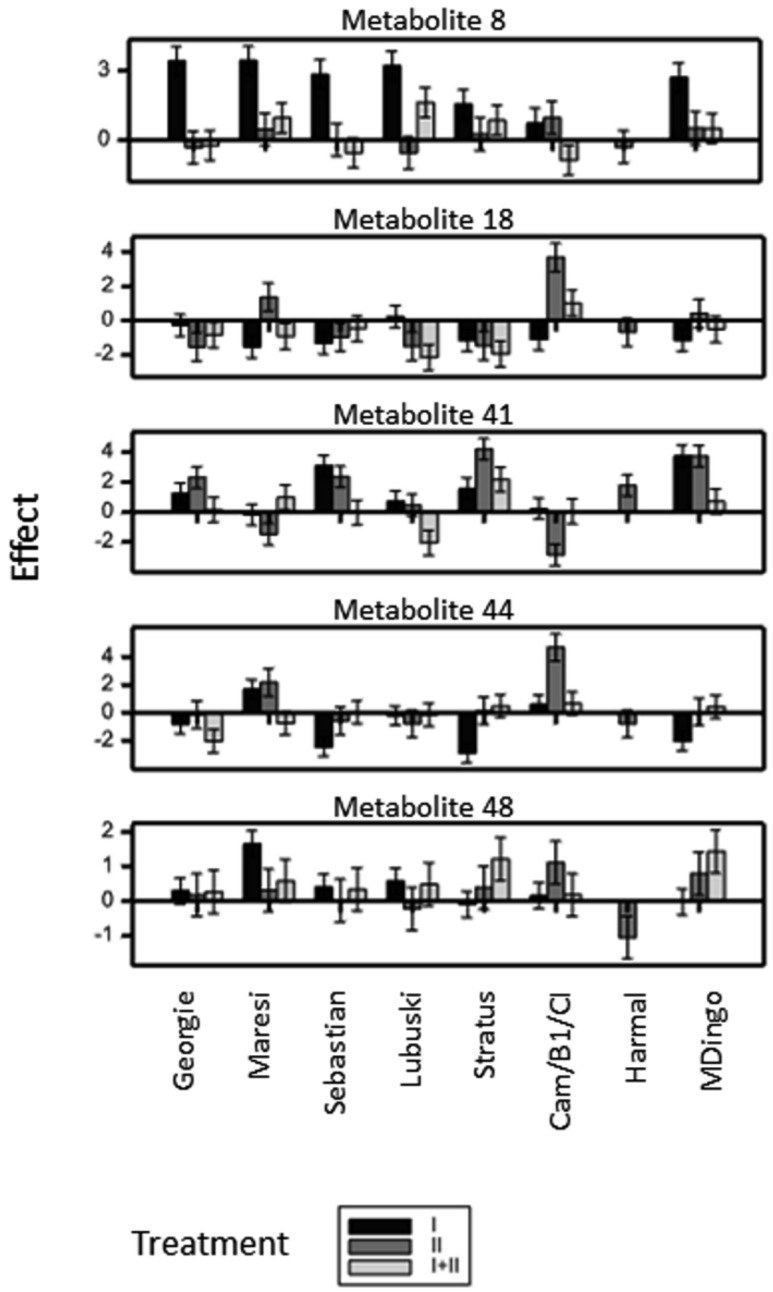
Variety-specific water deficiency effects observed on day 10 of treatment I, II and I+II, for identified metabolites with the effects significantly modified by variety and time under at least one treatment.

**Figure 4 ijms-21-06032-f004:**
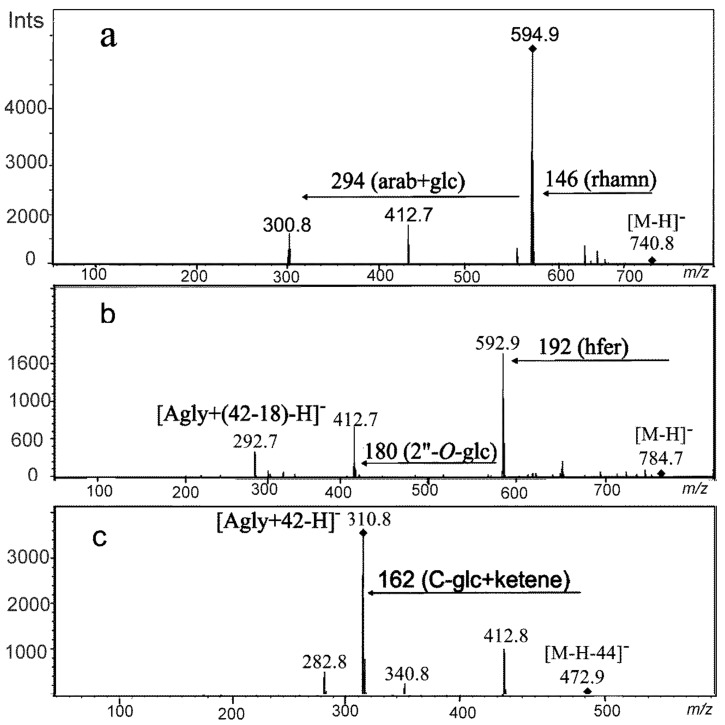
MS2 spectra in negative ionization of compounds (**a**) no. 117, quercetin *O*-rhamnoside-*O*-arabinosylglucoside; (**b**) no. 133, hydroxyferuloyl-isovitexin 2”-*O*-glucoside; (**c**) no. 152, isoitexin malonylated. (Agly-aglycon, glc-glucoside, rhamn-rhamnoside, hfer-hydroxyferulic acid).

**Figure 5 ijms-21-06032-f005:**
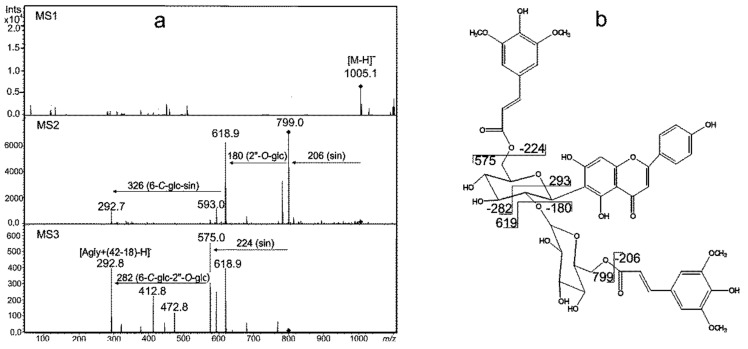
Mass spectra in negative ionization of compound 147, isovitexin 2”-O-glucoside disinapate (**a**) and putative fragmentation scheme (**b**) (Agly-flavone aglycon, glc-glucoside, sin-sinapic acid).

**Figure 6 ijms-21-06032-f006:**
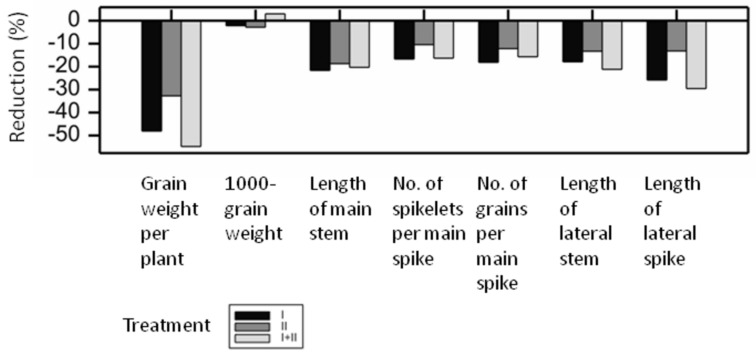
Mean effects of treatment I, II and I+II for phenotypic traits for which no variety-specific effects were observed (expressed in % of mean value under control conditions); for 1000-grain weight the effects were not significant.

**Figure 7 ijms-21-06032-f007:**
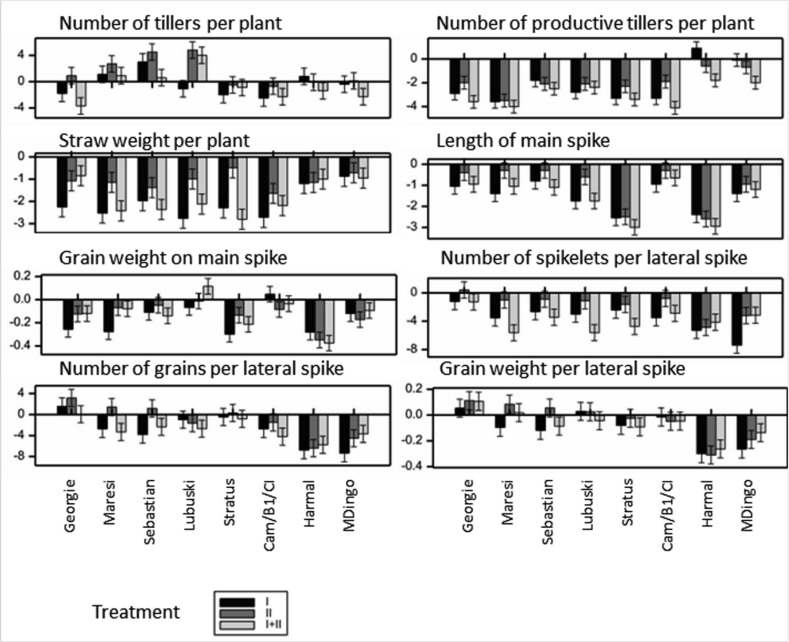
Effects of water deficiency under treatments I, II, I+II for phenotypic traits for which significant differences among varieties were observed in at least one treatment (*p <* 0.01).

**Figure 8 ijms-21-06032-f008:**
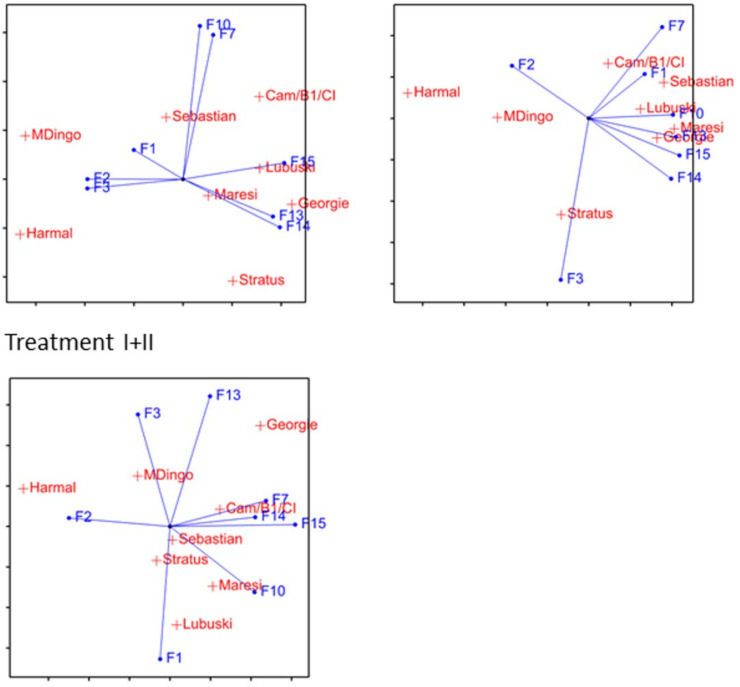
Biplots constructed for phenotypic traits with variety-specific water deficiency effects under treatments I, II, I+II.

**Figure 9 ijms-21-06032-f009:**
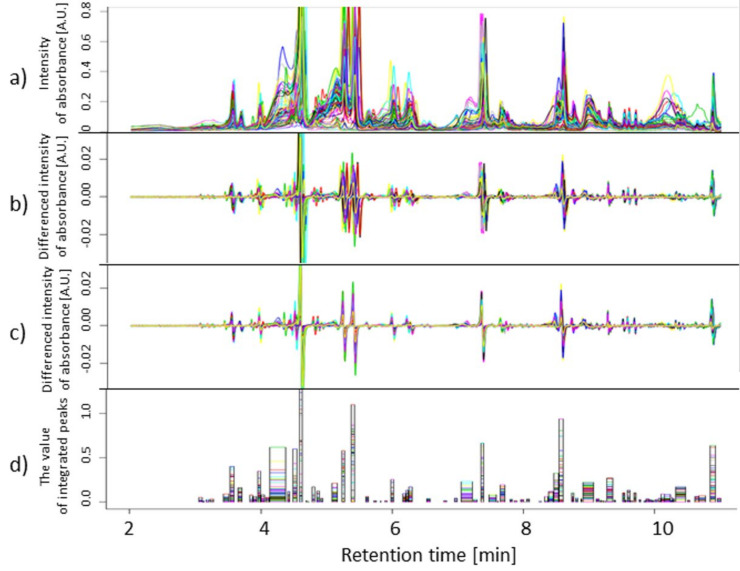
Preprocessing data from 62 chromatograms recorded at 330 nm for Maresi for all treatments and replications: (**a**) normalization by sample mass, (**b**) differentiation, (**c**) correlation optimized warping (COW), (**d**) peak detection. The last plot shows the width of peaks and their value after integration for individual chromatograms.

**Table 1 ijms-21-06032-t001:** Numbers of metabolites in model categories. Sign in brackets: large over (+) or under (-) representation of the category (test for independence using permutation, *p <* 0.001).

Drought	No Effect	D	VD	TD	VD,TD	VTD	Total
I	36 (-)	25	0	32	1	10 (+)	104
II	61	12 (-)	1	26	2	2	104
I+II	54	33 (+)	0	16	0	1	104

**Table 2 ijms-21-06032-t002:** Phenotypic traits of barley plants measured at maturity. Symbols D and VD indicate type of reaction of the trait to drought: D—mean drought effect; VD—variety-specific drought effect (ANOVA F-test, *p <* 0.01).

Trait Symbol	Group of Traits	Trait Name [units]	Treatment		
			I	II	I+II
F1	Tillers	Number of tillers per plant [pcs]	–	VD	VD
F2		Number of productive tillers per plant [pcs] ^a)^	VD	D	D
F3	Yield and biomass	Straw weight per plant [g]	VD	D	D
F4		Grain weight per plant [g]	D	D	D
F5		1000-grain weight [g]	–	–	–
F6	Main stem and spike	Length of main stem [cm]	D	D	D
F7		Length of main spike [cm]	D	VD	VD
F8		Number of spikelets on main spike [pcs]	D	D	D
F9		Number of grains on main spike [pcs]	D	D	D
F10		Grain weight on main spike [g]	D	D	VD
F11	Lateral stems and spikes	Length of lateral stems [cm]	D	D	D
F12		Length of lateral spikes [cm]	D	D	D
F13		Number of spikelets per lateral spike [pcs]	VD	D	D
F14		Number of grains per lateral spike [pcs]	VD	VD	D
F15		Grain weight per lateral spike [g]	VD	VD	-

^a)^ Number of tillers containing grains; [pcs]—pieces.

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
