# Peer review of "Phenolic Metabolites from Barley in Contribution to Phenome in soil Moisture Deficit"

_ijms, 2020, doi:10.3390/ijms21176032_

Round 1

Reviewer 1 Report

The introduction needs a bit more clarification as to why certain phenolic compounds were tested and their relevance to water/drought stress and water deficit. The research has merit but the findings and their significance are lost in clarity due to poor English. Additionally there were sections in the manuscript where the relevance of the compounds as a mediated response to water deficit was unclear. 

The authors would do well to look at the structure of the paper as there are numerous inconsistencies, in terms of Format and reference to their own tables/figures. For example; were the figures supplementary or part of the manuscript? If supplementary then the figures and tables wouldn't form part of the manuscript. Furthermore, the Material and Methods was out of place and was only found towards the end of the manuscript, after results and discussion which seemed out of place. Thus, I couldn't familiarise myself with the methodology and speculate as to the potential results; prior to reading the results and discussion.

In conclusion, the Manuscript seems rushed and incomplete, which is a pity.

Author Response

We are very grateful to the reviewer who has made a great effort in exposing the shortcomings and errors of that version. We have corrected the manuscript according to the remarks attached to the decision letter and below you can find our comments on the actions undertaken. For the readers’ convenience we have copied the text of the review (shown in black) and comment our response (shown in red). 

The introduction needs a bit more clarification as to why certain phenolic compounds were tested and their relevance to water/drought stress and water deficit.

Additional clarification and precise phenolic compounds description was added.

The research has merit but the findings and their significance are lost in clarity due to poor English. Additionally there were sections in the manuscript where the relevance of the compounds as a mediated response to water deficit was unclear. 

The text was sent to linguistic proofreading before submission. However, we additionally changed the content and meaning of unclear sentences and rebuilt the introduction and discussion. We made every effort to make the text clearer.

The authors would do well to look at the structure of the paper as there are numerous inconsistencies, in terms of Format and reference to their own tables/figures. For example; were the figures supplementary or part of the manuscript? If supplementary then the figures and tables wouldn't form part of the manuscript.

The numbering of tables and figures was changed. This mistake comes from the movement of the Materials and Methods section to the last part of the manuscript.

Furthermore, the Material and Methods was out of place and was only found towards the end of the manuscript, after results and discussion which seemed out of place. Thus, I couldn't familiarise myself with the methodology and speculate as to the potential results; prior to reading the results and discussion.

the Materials and Methods section was placed on the end of manuscript according to the journal guidelines.

In conclusion, the Manuscript seems rushed and incomplete, which is a pity.

Please find our correction and read the text once again carefully. The experimental setup is rather complex with three drought regimes and three/ four time points for eight barley varieties. Multidisciplinary approach can lead to misunderstanding and confusion in the results. We have made every effort to brighten up this complex system as much as possible and present the results in a simple and transparent way. This work is a continuation of a large project including also experiments with RIL lines of barley which was presented in

Piasecka, A.; Sawikowska, A.; Kuczyńska, A.; Ogrodowicz, P.; Mikołajczak, K.; Krystkowiak, K.; Gudyś, K.; Guzy‐Wróbelska, J.; Krajewski, P.; Kachlicki, P. Drought‐related secondary metabolites of barley (Hordeum vulgare L.) leaves and their metabolomic quantitative trait loci. Plant J. 2017, 89, 898-913. doi:10.1111/tpj.13430 .

We have made every effort to ensure that the work is written carefully and thoughtfully. We hope that the amendments made will change the way our work is viewed as rushed and incomplete.

Reviewer 2 Report

The manuscript explains a  study on the effect of drought stress at different development stages of barley varieties from different origins, and profiles of metabolites associated with the stresses compared to control condition. The experiment is well designed, and analysis is performed correctly. The manuscript meets the scope of International Journal of Molecular Sciences and it can be published once the following minor comments are considered.

Table S1_B: adding notation for I, II, K. What is K?

Why for each variety, K treatment has 7 time points

Table 2 is presented before table 1? I think it should be presented as Table 1 if it appears first.

For Table 1, there is no interaction between VxD detected to be significant? What does the shading refer to? Please explaining the shading in the caption.

Line 112, what does the “reactions” means? Reactions of what?

Figure 2. Some of the metabolites mentioned in the text from 134-151 (i.e. No. 9 and 48) were not in the Figure 2

Line 298: do all the traits look normally distributed? Please state this as if they are not, then transformation to data must be done.

Line 671-674:  from this text then readers would think D is treatment, V is variety, VD is the interaction between variety x treatment, TD is the interaction between time of treatment (T) and treatment (D). But in line 102-107, VD is variety-dependant class. Please change the acronym so that it is more consistent and less confusing for readers.

Line 112/156/190: group>class (please be consistent with wording, especially you also mentioned group ABCD later for time profile)

Figure 7. The y axis lacks label. Also, it is more meaningful and easier to express the effect of drought treatment if the authors change the scale from effect to percentage of reduction (ie 40% reduction in tiller number) as say reduction of 3 tillers or 3 g of TGW does not show the impact of drought treatment as expression of 20% or 50%.

Figure 9: Put in the labels for treatment I and II as there was label for only treatment I+II

Line 360-367: Where is the correlation data? This is actually an important and interesting dataset that needs to be presented (either as correlation matrix heatmap or supplemental table is fine).

Line: 424, 455, 458 and other: no period after for example “Fig.2”

Discussion section: is there any novel metabolite involved in water stress discovered in this manuscript that was not reported in the literature? If there is, please state as this would show how valuable this study is.

Line 556-559: while you summarize your finding, please include a concise summarizing sentence about the notable finding upon metabolites as it is the main focus of this manuscript.

Line 584: So all of 10 plants were measured individually and average was calculated? Why there were so many of plants in one pot? Usually with 2 reps x 3 plants per pots you would have six biological reps already. This high number of plants in one pot could pose another type of stress to the plants.

Line 585: there was no dry weight data. What do you mean?

Author Response

We are very grateful to the reviewer who has made a great effort in exposing the shortcomings and errors of that version. We have corrected the manuscript according to the remarks attached to the decision letter and below you can find our comments on the actions undertaken. For the readers’ convenience we have copied the text of the review (shown in black) and comment our response (shown in red). 

Table S1_B: adding notation for I, II, K. What is K?

Notation for treatment I, II and I+II as well as K was added. K is referred to control group of plants

Why for each variety, K treatment has 7 time points

Samples were collected from plants in treatments I, and II  and the corresponding controls (K) after 3, 6 and 10 days of moisture deficit, which is three time points and in treatment I+II and corresponding control (K) after 1, 3, 6 and 10 days of the second period of moisture deficit which is four time points. This differentiation may be a bit confusing, especially with a rather complicated set of experiments.

Table 2 is presented before table 1? I think it should be presented as Table 1 if it appears first.

The numbering of tables was changed. This mistake comes from the movement of the Materials and methods section to the last part of the manuscript

For Table 1, there is no interaction between VxD detected to be significant? What does the shading refer to? Please explaining the shading in the caption.

The shading is not required and was removed.

Line 112, what does the “reactions” means? Reactions of what?

The word “ reaction” referred to metabolomic changes in response to drought. For more precise description of results it was replaced with sentence: “Response to drought at metabolomic level” and in further text with “metabolomic changes”.

Figure 2. Some of the metabolites mentioned in the text from 134-151 (i.e. No. 9 and 48) were not in the Figure 2

Metabolite No 9 has not been mentioned in the part of text corresponding to Figure 2 (currently Fig. 1). The mention of metabolite 48 in this respect has been cancelled.

Line 298: do all the traits look normally distributed? Please state this as if they are not, then transformation to data must be done.

The observations were distributed symmetrically and bell-shaped, so that no transformation was necessary. This is stated now in line 560.

Line 671-674:  from this text then readers would think D is treatment, V is variety, VD is the interaction between variety x treatment, TD is the interaction between time of treatment (T) and treatment (D). But in line 102-107, VD is variety-dependant class. Please change the acronym so that it is more consistent and less confusing for readers.

In lines 102-107 the acronyms were changed according to the same description as in lines 671-647

Line 112/156/190: group>class (please be consistent with wording, especially you also mentioned group ABCD later for time profile)

The inconsistency was corrected by replacing the word “class” by “group” for names of ANOVA models and class was used for time-dependent profiles of metabolic changes.

Figure 7. The y axis lacks label. Also, it is more meaningful and easier to express the effect of drought treatment if the authors change the scale from effect to percentage of reduction (ie 40% reduction in tiller number) as say reduction of 3 tillers or 3 g of TGW does not show the impact of drought treatment as expression of 20% or 50%.

The figure was changed. As suggested by the reviewer, the effects are expressed now as percentage of mean value under control conditions, so that they represent reduction of the trait under three treatments (averaged over varieties). The y axis is labelled correspondingly. 

Figure 9: Put in the labels for treatment I and II as there was label for only treatment I+II

The labels were added

Line 360-367: Where is the correlation data? This is actually an important and interesting dataset that needs to be presented (either as correlation matrix heatmap or supplemental table is fine).

Pearson correlation coefficients are now presented in Supplementary Table S4. The table was not presented because, even though a number of correlations had large values, just a few were significant (at P < 0.01) - due to a low number of correlated effects (8). The correlations were more precisely described in new result section 2.5. Correlation between phenotypic and metabolomic traits in moisture deficit and discussed in new discussion section 3.4. correlation between phenotype and metabolome in moisture deficit

Line: 424, 455, 458 and other: no period after for example “Fig.2”

The mistake was corrected

Discussion section: is there any novel metabolite involved in water stress discovered in this manuscript that was not reported in the literature? If there is, please state as this would show how valuable this study is.

There was 11 new, drought-related metabolites 108, 117, 119, 124, 135, 146, 144, 146, 147, 148 and 153 which have not been previously described in barley. Notice about them was added at the end of the discussion section 3.2. However, this work is a continuation of the previous article on the RIL lineage (Piasecka et al. 2017) in which most of the interesting new metabolites have been described.

Line 556-559: while you summarize your finding, please include a concise summarizing sentence about the notable finding upon metabolites as it is the main focus of this manuscript.

A concise summarizing sentence about the notable finding upon metabolites was added

Line 584: So all of 10 plants were measured individually and average was calculated?

The experiment was described more clearly

 Why there were so many of plants in one pot? Usually with 2 reps x 3 plants per pots you would have six biological reps already. This high number of plants in one pot could pose another type of stress to the plants.

As you can see on graphical abstract there was really big pots used for experiments and 10 plants  have quite a lot of space. The metabolomic experiment was a part of multidisciplinary project and a large number of plants in one biological repetition (in one pot) resulted from the necessity to collect material for many other experiments.

Line 585: there was no dry weight data. What do you mean?

This sentence referred to the fact that measurements of some parameters like: straw weight per plant, grain weight per plant , 1000-grain weight were made on the dried material so as not to overestimate the results due to the water content. This sentence was changed to better understand the method.

Round 2

Reviewer 1 Report

Great improvement to the original manuscript.